# Identification and Comprehensive Structural and Functional Analyses of the *EXO70* Gene Family in Cotton

**DOI:** 10.3390/genes12101594

**Published:** 2021-10-09

**Authors:** Ya-Qian Zhu, Lu Qiu, Lu-Lu Liu, Lei Luo, Xin-Pei Han, Yao-Hua Zhai, Wen-Jing Wang, Mao-Zhi Ren, Ya-Di Xing

**Affiliations:** 1Zhengzhou Research Base, State Key Laboratory of Cotton Biology, School of Agricultural Sciences, Zhengzhou University, Zhengzhou 450001, China; zyqa1209@163.com (Y.-Q.Z.); qiulu@zzu.edu.cn (L.Q.); 202022582017482@gs.zzu.edu.cn (L.-L.L.); caoyutian163@163.com (L.L.); hnndhxp921230@163.com (X.-P.H.); zhaiyaohua999@163.com (Y.-H.Z.); wangwj@zu.edu.cn (W.-J.W.); 2State Key Laboratory of Cotton Biology, Institute of Cotton Research, Chinese Academy of Agricultural Sciences, Anyang 455000, China; 3School of Life Sciences, Zhengzhou University, Zhengzhou 450001, China

**Keywords:** *EXO70*, *Gossypium*, evolution analysis, transcriptome, expression analysis

## Abstract

The *EXO70* gene is a vital component of the exocytosis complex and participates in biological processes ranging from plant cell division to polar growth. There are many *EXO70* genes in plants and their functions are extensive, but little is known about the *EXO70* gene family in cotton. Here, we analyzed four cotton sequence databases, identified 165 *EXO70* genes, and divided them into eight subgroups (*EXO70A*–*EXO70H*) based on their phylogenetic relationships. *EXO70A* had the most exons (≥11), whereas the other seven each had only one or two exons. Hence, *EXO70A* may have many important functions. The 84 *EXO70* genes in Asian and upland cotton were expressed in the roots, stems, leaves, flowers, fibers, and/or ovules. Full-length *GhEXO70A1-A* cDNA was homologously cloned from upland cotton (*Gossypium hirsutum*, *G. hirsutum*). Subcellular analysis revealed that GhEXO70A1-A protein was localized to the plasma membrane. A yeast two-hybrid assay revealed that GhEXO70A1-A interacted with GhEXO84A, GhEXO84B, and GhEXO84C. *GhEXO70A1-A* silencing significantly altered over 4000 genes and changed several signaling pathways related to metabolism. Thus, the *EXO70* gene plays critical roles in the physiological functions of cotton.

## 1. Introduction

Vesicle transport is an extremely important cytological process in eukaryotes. It moves proteins, lipids, and other substances between the inner membrane system and the cells, and establishes cell polarity, secretion, growth, division, and wall formation [1]. Tethering is a key step in vesicle transport. Large multi-subunit tethering complexes were first discovered in yeast [2]. Exocysts tether different vesicles to the exocytosis site required for cellular secretion [3]. They are evolutionarily conserved octameric protein complexes composed of Sec3, Sec5, Sec6, Sec8, Sec10, Sec15, EXO70, and EXO84 [4,5]. EXO70 plays a key role in exocyst assembly [6]. It recruits exocysts on the target membrane and interacts with Rho protein to regulate SNARE complex assembly and activation there via SEC6. In this manner, EXO70 mediates polar exocytosis [7,8].

The exocyst subunits are encoded by a single gene in yeast and just a few genes in metazoans. However, 23 EXO70 subunits encoded by various loci have been identified in Arabidopsis [9,10]. The *EXO70s* in terrestrial plant genomes have even more copies. This phenomenon is unique to the EXO70 subunit of the exocyst [11]. In the fungal and animal genomes sequenced to date, only one *EXO70* coding gene was found. Hence, multiple *EXO70* gene copies are unique to higher terrestrial plants [12]. Certain EXO70 functions might have been alienated during evolution and participated in other biological processes besides membrane vesicle transport. Alternatively, various EXO70 functions are specialized and form different exocysts from other subunits that participate in specific membrane vesicle transport processes in the organization, carrier substrate, or transport link [12]. The expression profiles of the 23 members of the Arabidopsis *EXO70* family have been analyzed. Expression of this gene family has the following characteristics: spatiotemporal expression specificity at the cell and tissue levels; no constitutive expression; and specific expression in dividing, growing, differentiating, and secreting cells [12]. Plant *EXO70* gene family members participate at the transcriptional level in the biological processes of different cell types via cell- and tissue-specific expression patterns.

EXO70 is an important part of the secretory complex mediating exocytosis, and it regulates neurite growth in animal cells, epithelial cell polarity, and cell movement and morphogenesis [13,14,15,16,17]. In plants, EXO70 regulates pollen tube elongation and polarization, root hair growth, cell wall material deposition, cell plate activation and maturation, defense and autophagy, and so on [10,18,19,20,21]. Defects in *AtEXO70C2* gene function affect pollen tube growth, which results in significant male-specific transmission defects in Arabidopsis [22]. EXO70H4- and PMR4-dependent corpus callosum deposition in trichomes is necessary for cell wall silicification [23]. *EXO70B1* knockdown resulted in impaired light-induced stomatal opening [24]. *AtEXO70B1* and *AtEXO70B2* regulate *FLS2* to participate in plant immune response [25]. AtEXO70D regulates cytokinin sensitivity by mediating the selective autophagy of Type-A ARR protein, thereby maintaining cell homeostasis and normal plant growth and development [26]. OsEXO70A1, OsEXO70L2, and AtEXO70A1 affect tracheary element (TE) development [27,28,29]. Hence, the roles of EXO70 in plant organ development have undergone differentiation.

Cotton is a major global economic crop. It is a source of seed, fiber, oil, and medicine [30,31]. The development of novel high-quality cotton varieties is of great commercial importance. The high copy numbers and tissue-specific functions of the *EXO70* gene in plants suggest that targeting EXO70 to construct high-quality cotton is feasible. To date, however, few studies have investigated the cotton *EXO70* gene. The only report is that GhEXO70B1 may respond to stressors by mediating cell autophagy [32]. Here, we conducted evolutionary analyses, systematically named cotton *EXO70* gene family members, and examined the functions of GhEXO70A1-A. It is believed that the discoveries of the present work will provide theoretical and empirical references for future research on cotton EXO70.

## 2. Materials and Methods

### 2.1. Identification of EXO70 Family Members in Cotton

Upland cotton (*G. hirsutum*, NAU), sea island cotton (*Gossypium barbadense*, *G. barbadense*, HAU), their common ancestor Asian cotton (*Gossypium arboretum*, *G. arboretum*, CRI), and Raymond cotton (*Gossypium raimondii*, *G. raimondii*, JGI) genomes and gene structure annotation file were downloaded from the CottonFGD genome database (https://cottonfgd.org/, accessed on 15 December 2020). The tobacco genome and gene structure annotation file were downloaded from the Ensembl website (http://ensembl.org/index.html, accessed on 14 May 2021). The Arabidopsis EXO70 protein sequences were obtained from the TARE website (https://www.arabidopsis.org/, accessed on 12 August 2020). The CDS sequence from the downloaded genome and gene structure annotation file was extracted, and then translated into amino acid sequences. A local BLASTP search was completed to identify complete *EXO70* genes, using A. thaliana EXO70 sequences as queries. A BLASTP search (e value: 1e−5) was used to obtain a dataset of EXO70 proteins. To identify all members of the *EXO70* gene family, previous *EXO70* gene sequences were analyzed using the Simple Modular Architecture Research Tool (NCBI CDD; https://www.ncbi.nlm.nih.gov/cdd/, accessed on 15 August 2020) and Pfam (http://pfam.janelia.org/, accessed on 15 August 2020). The *EXO70* genes were annotated according to their corresponding orthologs in Arabidopsis and their chromosomal positions in cotton. The genes in *G. hirsutum* and *G. barbadense* were named according to their homologs in each subgenome, where “A” and “D” represent the At and Dt subgenomes, respectively [33]. Assignment of each gene to the At subgenome or Dt subgenome was based on its DNA sequence homology with an A genome diploid species (*G. arboretum*) or a D genome diploid species (*G. raimondii*). The gene positions of *GaEXO70*, *GrEXO70*, *GhEXO70*, and *GbEXO70* on the chromosome were determined from the genome data and were displayed using Tbtools software (https://www.tbtools.com/user-guide/installation, accessed on 20 August 2020). ExPASy software (https://www.expasy.org, accessed on 10 October 2020) was used to predict the theoretical molecular weights (MW) and isoelectric points (pI) [34]. The subcellular location of the EXO70 protein was established via the Cell-PLoc 2.0 (http://www.csbio.sjtu.edu.cn/bioinf/euk-multi-2/, accessed on 20 October 2020). The 2-kb region upstream of the *EXO70* start codon was found on the CottonFGD website and submitted to the PlantCARE database (http://bioinformatics.psb.ugent.be/webtools/plantcare/html/, accessed on 10 November 2020) to localize *cis*-acting elements [35].

### 2.2. Phylogeny, Gene Structure, Conserved Domains, and Motif Analysis

The muscle program in MEGA-X software (https://www.megasoftware.net/dload_win_gui, accessed on 11 November 2020) was used to align multiple amino acid sequences. The maximum likelihood (ML) method and Jones–Taylor–Thornton (JTT) model were used to construct a rootless phylogenetic tree through 1000 repeated bootstrap tests [36]. The phylogenetic tree was visualized with the iTOL tool (https://itol.embl.de, accessed on 15 November 2020) [37]. The *EXO70* gene structure was obtained from the genome dataset and displayed with Tbtools [38]. The EXO70 protein conserved domain was sought through the NCBI conserved domain database (CDD) (https://www.ncbi.nlm.nih.gov/cdd/, accessed on 12 November 2020) and displayed with Tbtools [38]. The EXO70 amino acid sequence in the *G. hirsutum* protein dataset was submitted to the MEME program (https://meme-suite.org/meme/, accessed on 18 November 2020) to identify conserved motifs [39]. To determine their evolutionary relationships and predict the EXO70 protein TM structure, the TMHMM-2.0 website (http://www.cbs.dtu.dk/services/TMHMM/#opennewwindow, accessed on 25 November 2020) was consulted.

### 2.3. Expression Pattern Analysis

The gene transcription level was calculated based on publicly released data. A *GaEXO70* gene expression dataset was obtained from Cotton FGD (https://cottonfgd.org/, accessed on 10 January 2021). RNA-Seq datasets for various *G. arboretum* tissues were procured from the BioProGene project (https://bioprogene.com, accessed on 15 January 2021; Accession No. PRJNA594268). The *G. hirsutum* gene expression dataset was acquired from CottonGen (https://www.cottongen.org/, accessed on 20 January 2021). HISAT (https://ccb.jhu.edu/software/hisat/index.shtml, accessed on 23 January 2021) and StringTie (https://ccb.jhu.edu/software/stringtie/, accessed on 23 January 2021) software tools were used to estimate the gene transcription levels as million fragments per kilobase (FPKM) values for heatmap drawing.

A total RNA extraction kit for plant polysaccharides and polyphenols (No. DP441; Tiangen Biotech, Beijing, China) was used to extract total RNA from roots, stems, leaves, flowers, cotyledons, and ovules at the 0 DPA, 10 DPA, and 20 DPA stages of upland cotton. A reverse transcription kit (No. E047; Novoprotein Scientific Inc., Summit, NJ, USA) was used according to the manufacturer’s instructions to reverse transcribe the extracted RNA into cDNA. Quantitative *GhEXO70A1-A* primers (Appendix A) were designed, and qRT-PCR was performed to analyze relative *GhEXO70A1-A* expression in different cotton tissues.

### 2.4. Cloning of GhEXO70A1-A Gene

*AtEXO70A1* homologous genes were sought in CottonFGD. Vector NTI Advance (Thermo Fisher Scientific, Waltham, MA, USA) was used to compare the downloaded sequences. The group A genes with the highest homology were named *GhEXO70A1-A*. Primers were designed using Vector NTI Advance software according to the *GhEXO70A1-A* gene coding sequence (Appendix A). The cDNA was amplified by PCR to obtain the *GhEXO70A1-A* gene, and the PCR product was purified and recovered. It was cloned into the B-zero vector, and it was sent out for sequencing (Sangon Biotech (Shanghai, China)).

### 2.5. Real-Time Quantitative PCR (RT-qPCR) Analysis

NovoStart SYBR qPCR SuperMix Plus (E096; Novoprotein) was used to perform qRT-PCR detection of the foregoing cDNA. *GhUBQ7* was the internal reference gene. The quantitative primer statistics are shown in Appendix A. qRT-PCR was performed in a LightCycler 480 Ⅱ (Roche Diagnostics, Basel, Switzerland). The parameters of the PCR system and procedures were set according to the quantitative fluorescence kit instructions. There were three biological replicates per reaction. Relative gene expression was calculated by the 2^−△△Ct^ method [40]. *GhEXO70A1-A* gene expression patterns were analyzed in different cotton tissues.

### 2.6. Subcellular Localization Detection of GhEXO70A1-A

The *GhEXO70A1-A* coding region sequence was cloned into a 2300-YFP vector to obtain the p35S-GhEXO70A1-A-YFP plasmid (the primers are listed in Appendix A), which was transformed into an Agrobacterium GV3101 competent strain (No. AC1001; Shanghai Weidi Biotechnology). A single clone was selected to identify the positive strain. Expanded culture was performed to inject tobacco leaves, which were then incubated in the dark at 28 °C for 16 h, illuminated for 48 h, and observed under a confocal laser microscope (No. FV1200; OLYMPUS, Tokyo, Japan). Green fluorescent protein signals in the tobacco leaf epidermal cells were photographed. The 2300-YFP empty vector was the control, and there were three independent biological replicates.

### 2.7. Yeast Double-Hybrid Assay

*Eco*RI and *Bam*HI were used to digest the pGADT7 and pGBKT7 vectors. Homologous recombination was used to clone the *GhEXO70A1-A* gene into the pGBKT7 vector (C115, Vazyme Biotech co., Ltd.). The *GhEXO84A*, *GhEXO84B*, *GhEXO84C*, *GhSEC5*, *GhSEC6*, *GhSEC8*, *GhSEC10*, *GhSEC15A*, and *GhSEC15B* genes were cloned into the pGADT7 vector. The plasmids were co-transformed into a yeast Y2H competent strain (No. YC1002; Hangzhou Yizhi Biotechnology Co. Ltd., Hangzhou, China), spread onto SD-Trp/-Leu medium, and cultured in a 30 °C incubator. Single clones were selected, diluted with sterile water, and applied to SD-Trp/-Leu/-His/-Ade (20 mg/mL X-gal plus 1 mg/mL AbA). The primers are listed in Appendix A.

### 2.8. Virus-Induced GhEXO70A1-A Gene Silencing

An online website (http://vigs.solgenomics.net/, accessed on 20 March 2021) was used to determine the *GhEXO70A1-A* silencing sequence. Primers were designed for PCR amplification and ligated to the pCLCRVA vector to obtain the pCLCRV–GhEXO70A1-A recombinant plasmid. The pCLCRV–GhEXO70A1-A, pCLCRVB, and pCLCRVA plasmids were transformed into *Agrobacterium tumefaciens* GV3101, and the positive clones were selected. The pCLCRV–GhEXO70A1-A and pCLCRVA plasmids were mixed with equal amounts of pCLCRVB resuspension. After 3 h, a sterile syringe was used to inject the mixture into 14-d cotton seedling cotyledons. A CLCRV empty-load injection was the control (CK). After the positive seedling leaf yellowing phenotype appeared, total RNA was extracted and *GhEXO70A1-A* gene expression was determined by quantitative fluorescence PCR detection. The quantitative primers are listed in Appendix A.

### 2.9. Transcriptome Analysis

Based on the results of the *GhEXO70A1-A* gene silencing expression analysis, total RNA was extracted from the fifth and sixth leaves of the virus-induced *GhEXO70A1-A* gene-silenced and control plants. The samples were subjected to gel electrophoresis, and transcriptome sequencing was performed at Beijing Nuohe Zhiyuan Technology Co. Ltd. (Beijing, China).

## 3. Results

### 3.1. Identification and Analysis of the Phylogenetic Relationship of the EXO70 Gene Family in Cotton

The exocytosis complex subunits comprise mostly *EXO70* gene family members. There are 23 and 47 *EXO70* genes in the model dicotyledon *Arabidopsis thaliana* and the monocotyledon rice, respectively. Here, we identified 165 *EXO70* genes among the four cotton subspecies included in the CottonFGD database, namely, *G. hirsutum*, *G. barbadense*, *G. arboretum*, and *G. raimondii*. There were 27, 26, 55, and 57 genes in *G. arboretum*, *G. raimondii*, *G. barbadense*, and *G. hirsutum*, respectively. We also identified 48 *EXO70* genes in the tobacco database (Figure 1A). A phylogenetic analysis of the evolutionary relationships of 23 Arabidopsis *EXO70s*, 41 rice *EXO70s*, 165 cotton *EXO70s*, and 48 tobacco *EXO70s* (Figure 1C) showed that cotton *EXO70* resembled Arabidopsis *EXO70*. Both of these plants are dicotyledons [9,41] and their *EXO70s* could be divided into eight categories. Relative to the monocotyledon rice, the dicotyledons lacked four *EXO70* categories, such as EXO70I–EXO70L [42]. Hence, the *EXO70* gene may be markedly differentiated between monocotyledons and dicotyledons. Monocotyledons possess more *EXO70* genes than dicotyledons. Based on the phylogenetic tree, the grouping and naming of Arabidopsis, cotton *EXO70s* can be divided into eight subgroups (EXO70A–EXO70H) containing 12, 6, 29, 12, 27, 12, 22, and 45 genes, respectively (Figure 1B).

According to the cotton *EXO70* gene classification, we named the 57 *EXO70* genes in upland cotton as GhEXO70A1–GhEXO70A2, GhEXO70B, GhEXO70C1–GhEXO70C5, GhEXO70D1–GhEXO70D2, GhEXO70E1–GhEXO70E6, GhEXO70F1–GhEXO70F2, GhEXO70G1–GhEXO70G4, and GhEXO70H1–GhEXO70H8. Groups A and D were represented by -A and -D, respectively. We predicted their genome locations, protein lengths, numbers of exons, isoelectric points, protein molecular weights, and subcellular locations. The numbers of exons widely varied among *GhEXO70* genes. All four *GhEXO70A* genes had the most exons (≥11 each) (Table 1). Further analysis of the exons of the *EXO70* gene in Arabidopsis and rice showed that only Group A contained more exons in Arabidopsis and rice. The number of *EXO70* in Arabidopsis A group was ≥9, and the number of *EXO70* in rice A group was ≥12. Therefore, this phenomenon is not unique to cotton *EXO70s*, but is conserved in plants (Appendix A). The subcellular localization prediction results showed that GhEXO70 was mostly localized in the cell membrane, cytoplasm, or nucleus, which was consistent with the reported subcellular localization results of EXO70 from H. villosa [43] (Table 1).

### 3.2. Chromosome Distribution Analysis of EXO70 in the Cotton Genome

Diploid *G. arboretum*, *G. raimondii*, and Arabidopsis have 27, 26, and 23 *EXO70* genes, respectively, while diploid rice has 47. Tetraploid *G. hirsutum* and *G. barbadense* have 57 and 55 *EXO70* genes, respectively. The 27 *EXO70* genes of *G. arboretum* are located on chromosomes 1–2, 4–5, 7, and 9–13, respectively (Figure 2C). The 26 *EXO70* genes of *G. raimondii* are located on chromosomes 1–3 and 6–12, respectively (Figure 2D). The 57 *EXO70* genes of *G. hirsutum* are located on chromosomes 1, 3–5, 7, and 9–13 in groups A and D, respectively (Figure 2A). The 55 *EXO70* genes of *G. barbadense* are located on chromosomes 1, 3–5, 7, and 9–13 in group A and on chromosomes 1, 3–5, 7, 9–10, and 12–13 in group D (Figure 2B).

Statistical analysis of the *EXO70* gene distributions on the chromosomes revealed that there were relatively more *EXO70* genes on chromosomes 5 and 9 in *G. arboretum*, *G. barbadense*, and *G. hirsutum*, but no *EXO70* genes on chromosome 6 or 8. The *EXO70* gene on chromosome 9 was distributed in *G. raimondii*, but that which was on chromosome 5 was not distributed. The *EXO70* gene distributions on chromosomes 6 and 8 of *G. raimondii* (four and two, respectively) were the opposite of those for the other three cotton species (Figure 2; Table 2).

The number of *EXO70* genes in tetraploid cotton was nearly twice that in diploid cotton. The diploid cotton species (*G. arboretum* and *G. raimondii*) contained two EXO70As, one EXO70B, two EXO70Ds, and two EXO70Fs, whereas the tetraploid cotton species had twice these *EXO70* gene copy numbers (Table 3). The numbers of EXO70Cs, EXO70Gs, and EXO70Hs in tetraploid cotton were twice those in the autodiploid species and equal to the sum of the number in the allodiploid species (Table 3). In polyploid cotton, then, the number of *EXO70* genes increases via genome polyploidization. Most *GhEXO70* genes are highly parallel in the At group and Dt subgenome. The exception is that GhEXO70E2-D and GhEXO70E4-D have no homologs in the At subgenome, while GhEXO70H8-A has no homologs in the Dt subgenome, indicating that they may be lost during evolution.

### 3.3. Analysis of EXO70 Gene Structure in G. hirsutum

*G. hirsutum* is the major global cotton variety and was the focus of research attention here. Structural analysis of its 57 *GhEXO70* genes showed that all of them had one or two exons except for *GhEXO70A*, which had 10 or 11 exons. All *GhEXO70* genes with similar structures are grouped in the same clade. Moreover, the genes with closely related phylogeny in the same subgroup also had similar structures. Within the same subgroup, however, certain genes exhibited entirely different structures. *GhEXO70E2-D* contained two exons, while the other genes within the same subgroup had only one. Similarly, *GhEXO70G4* contained two exons, whereas *GhEXO70G1*–*GhEXO70G3* each contained a single exon. *GhEXO70H2-A* and *GhEXO70H8-A* each contained two exons while the other genes within the same subgroup had only one (Figure 3).

We used MEME online software to analyze the conserved motifs in the GhEXO70 protein and study its motif composition diversity and conservation. Figure 3 shows that 10 motifs (1–10) were identified, and each one was localized mainly to the *C*-terminal of the gene. Therefore, the *C*-terminal sequence of the GhEXO70 protein is highly conserved. The motif types revealed that the *GhEXO70* gene members in subgroups A, B, C, and D were highly conserved and included all motifs. The *GhEXO70s* gene members in the other subgroups presented with obvious differences in motif type distribution, and some of them were lost. GhEXO70E2-D, GhEXO70E4-D, GhEXO70H1-D, and GhEXO70G2-D contained two, three, five, and six motifs, respectively. The functions of Motifs 1–10 have not been elucidated. Nevertheless, analysis of the conserved domains via the NCBI Conserved Domain Database (CCD) disclosed that they comprise the Exo70 domain (Figure 3).

The PFam03081 domain at the *C*-terminus of the EXO70 protein is characteristic of the EXO70 superfamily [12], and all 165 predicted homologous clone EXO70 proteins possess it. However, the amino acid sequence lengths differed among EXO70 proteins and were in the range of 134–735 aa (average length = 618.736842105263 aa) (Table 1). It was discovered that most *GhEXO70* genes lacked a transmembrane (TM) structure. Only *GhEXO70E2-D* might possess a transmembrane region. Therefore, it may have evolved along with eukaryote evolution (Appendix A). For the prediction of the transmembrane domain of Arabidopsis EXO70, the results showed that AtEXO70C1, AtEXO70C2, AtEXO70H5, AtEXO70H8, and AtEXO70A3 have transmembrane domains, but they are not obvious, and the other EXO70s have no transmembrane domains (Appendix A). The prediction results of rice EXO70 show that OsEXO70A3, OsEXO70A4, OsEXO70H1a, OsEXO70H1b, OsEXO70H2, OsEXO70H3, OsEXO70H4, OsEXO70I3, OsEXO70I4, OsEXO70L1, OsEXO6K1, OsEXO70L, OsEXO70J1, OsEXO70J1, OsEXO70J2, OsEXO70J6, OsEXO70J8, OsEXO70K1, OsEXO7K2, and OsEXO70L1 have a transmembrane domain. In addition, OsEXO70A4 has a more obvious transmembrane domain at the C-terminus, and none of the other rice EXO70s has a transmembrane domain (Appendix A). Among the prediction results of the transmembrane domain of cotton EXO70, only GhEXO70E2-D has a transmembrane domain, and the others have no transmembrane domain. Both Arabidopsis and cotton contain fewer EXO70s with transmembrane domains. As rice is a monocot, it may be evolving to have more EXO70, and there are more EXO70s with transmembrane domains.

### 3.4. Analysis of EXO70 Gene Expression Patterns in G. arboretum and G. hirsutum

Gene expression has spatiotemporal properties. The expression patterns of the various members of the *EXO70* gene family may indicate the potential biological effects of these genes. We analyzed expression profile data in the CottonFGD and Cottongen (https://www.cottongen.org/, accessed on 22 March 2021) databases to clarify the spatiotemporal expression characteristics of the *EXO70* gene. In *G. hirsutum* and *G. arboretum*, the *EXO70* gene is commonly expressed in the roots, stems, leaves, flowers, fibers, and ovules and has spatiotemporal properties (Figure 4 and Appendix A). *GhEXO70A1-A*, *GhEXO70A1-D*, *GhEXO70B-A*, *GhEXO70B-D*, *GhEXO70D1-A*, *GhEXO70E1-A*, *GhEXO70E6-A*, *GhEXO70F2-D*, and other genes in *G. hirsutum* are generally expressed at high levels and in various tissues. The *GhEXO70H3-A* gene is expressed mainly in the stamens, whereas the *GhEXO70H5-A* and *GhEXO70H5-D* genes are expressed mainly during the early stages of ovule development. In *G. arboretum*, the *GaEXO70A1*, *GaEXO70E4*, *GaEXO70B*, *GaEXO70F1*, *GaEXO70F2*, *GaEXO70D1*, *GaEXO70E1* genes are generally highly expressed in different tissues, while *GaEXO70A2* is expressed mainly in the 15D fibers (Appendix A).

Ubiquitous *EXO70* expression suggests that this gene is implicated in cotton growth and development. The *GaEXO70A2* gene is expressed mainly in the fibers and might participate in cotton fiber development. The *GhEXO70H3-A* gene is expressed mainly in the stamens and may be associated with cotton fertility. The *GhEXO70H5-A* and *GhEXO70H5-D* genes are expressed mainly in the early stages of ovule development and could be involved in cotton seed formation.

### 3.5. EXO70 Gene Transcription Regulation Analysis

Spatiotemporal gene expression is regulated mainly by transcription factors (TFs) and epigenetics [44]. The observed differences in spatiotemporal expression of the various *EXO70* genes may be related to their promoter specificity. We intercepted the 2-kb sequence upstream of the cotton *EXO70* gene start codon and used the PlantCARE database (http://bioinformatics.psb.ugent.be/webtools/plantcare/html/, accessed on 20 April 2021) to analyze the *cis*-elements in the promoter region. A total of 1081 *cis*-elements were predicted in the 57 *GhEXO70* gene promoter regions. Of these, 10 and 11 categories were related to phytohormones and environmental stressors, respectively. The functions of the *cis*-elements in phytohormone and environmental stress response are highlighted in Figure 5. Among the predicted phytohormone response elements, the ERE, ABRE, and CGTCA motifs were the most abundant. Hence, the *GhEXO70* gene might respond to ethylene, abscisic acid, and methyl jasmonate (MeJA) (Figure 5A). Ten environmental stress-related elements were identified and mainly involved drought stress (MYC), stress response (STRE), and anaerobic induction (ARE) (Figure 5B). Therefore, the *EXO70* gene may participate in the response to adversity. To further verify whether the above cis-acting elements are unique to cotton *EXO70s*, we also analyzed the *EXO70s* gene promoters in Arabidopsis and rice. The results indicate that the promoters of *EXO70* genes in Arabidopsis and rice also contain cis-acting elements that respond to environmental stress and plant hormones (Appendix A). It shows that this phenomenon is not unique to cotton *EXO70*, but is conserved in plants.

### 3.6. Expression Analysis and Subcellular Location of GhEXO70A1-A

The *EXO70A1* gene is the most widely studied of all plant EXO70 genes. In Arabidopsis, AtEXO70A1 differentiates tubular molecules and regulates seed coat, root hair, stigma papillae development, and Kjeldahl band formation [45,46,47]. OsEXO70A1 plays important roles in vascular bundle differentiation and mineral nutrient assimilation [28]. In this study, we used GhEXO70A1-A in an experimental study on cotton *EXO70* genes. We tested the *GhEXO70A1-A* gene expression patterns. *GhEXO70A1-A* was predominantly expressed in the stems, leaves, and flowers but its expression levels were low in the roots, ovules, and cotyledons (Figure 6A).

Subcellular GhEXO70A1-A protein localization predicted its roles in biological processes. Transient 35S-GhEXO70A1-A-GFP expression in tobacco produced a fluorescent signal. GhEXO70A1-A induced signals on the plasma membrane (Figure 6B). Thus, GhEXO70A1-A was localized to the endomembrane system. This discovery was consistent with the roles of EXO70s in vesicle transport.

### 3.7. GhEXO70A1-A Protein Interaction Analysis

We used a yeast two-hybrid (Y2H) assay to explore the interactions among GhEXO70A1-A and the other subunits of the exocytosis complex. Plasmids containing GhEXO70A1-A and the other subunits of the exocytosis complex were co-transformed into Y2H Gold cells, which can grow on SD/-Leu-Trp. However, the cells were inoculated onto SD/-Ade/-His/-Leu/-Trp medium and only GhEXO70A1-A and GhEXO84A, Gh EXO84B, GhEXO84C co-transformed cells could grow on it and express X-α-Gal activity. GhEXO70A1-A interacted with EXO84A, EXO84B, and EXO84C (Figure 7), which means that it may function as a subunit of the exocytosis complex.

### 3.8. VIGS Silencing of GhEXO70A1-A Causes Changes in Signaling Pathways and Gene Expression

Gene silencing is an effective method of studying gene function. To explore the functions of GhEXO70A1-A in cotton, we constructed *GhEXO70A1-A*-gene-silenced cotton plants by virus-induced gene silencing (VIGS). qPCR demonstrated that the *GhEXO70A1-A* gene was successfully knocked down (Figure 8A). We then used next-generation sequencing (NGS) technology to detect any changes in the transcriptome of *GhEXO70A1-A*-silenced leaves. We sorted out the expression of other EXO70 genes in the NGS data, and the results are shown in Appendix A; GhEXO70B-D, GhEXO70B-A, GhEXO70E6-D, GhEXO70F1-D, GhEXO70E3-D, GhEXO70H6-D, GhEXO70E1-D, GhEXO70H6-A, GhEXO70H3-D, GhEXO70D1-D, GhEXO70E3-A, GhEXO70C1-A, GhEXO70E5-A, and GhEXO70C5-A have significant changes, and there are no significant changes in other EXO70 genes. However, except for GhEXO70C1-A, GhEXO70H6-D, and GhEXO70H6-A, which decreased to 32.1%, 46.9%, and 56.6% of the control, all other genes fell to more than 60% of the control, and the fold increase was also less than 1. Although the three genes GhEXO70C1-A, GhEXO70H6-D, and GhEXO70H6-A declined slightly, their expression abundance was also very low. The above results show that the knockdown of GhEXO70A1-A by VIGS does affect the expression of other EXO70 genes, but the effect is not significant after analysis. The changes in differential genes should be mainly caused by the changes in GhEXO70A1-A.

Correlation analyses among samples disclosed significant differences between the *GhEXO70A1-A*-silenced (EXO70A1) and the control (VIGS-CK) groups (Figure 8B). Thus, *GhEXO70A1-A* silencing in cotton altered the gene expression profiles. Differentially expressed genes (DEG) were those that met the criteria of |log2(Fold Change)| ≥ 1 and *p* ≤ 0.05. A total of 3264 upregulated and 1103 downregulated genes were screened, as shown in a volcano graph (Figure 8C) and a heat map (Figure 8D). Kyoto Encyclopedia of Genes and Genomes (KEGG) functional enrichment of the DEGs (Figure 8E) displayed 13 pathways with *p* < 0.01. These included photosynthesis antenna protein, phenylpropane biosynthesis, flavonoid biosynthesis, starch and sucrose metabolism, circadian rhythm—plant, keratin, cork and wax biosynthesis, steroid biosynthesis, sesquiterpenoid and triterpenoid biosynthesis, glutathione metabolism, cyano-amino acid metabolism, photosynthesis, and glucosinolate biosynthesis (Table 4). GSEA results showed that GhEXO70A1-A was significantly related to photosynthesis antenna protein, photosynthesis, and circadian rhythm—plants (Figure 8F–H). Of the 13 significantly different pathways, all except for circadian rhythm—plants were related to metabolism. Therefore, cotton leaf GhEXO70A1-A may regulate biochemical anabolism and catabolism.

## 4. Discussion

### 4.1. Evolutionary Relationships of the Cotton EXO70 Gene Family

The evolutionary relationships of the *EXO70* gene family in Asian, Raymond, upland, and sea island cotton were inferred according to their total numbers, classifications, chromosome distributions, and structures. Allotetraploid cotton was crossed from its ancestors *G. raimondii* and *G. arboretum* about 5 million years ago [48]. There are about twice as many *EXO70* genes in tetraploid cotton (57 and 55, respectively) as there are in diploid cotton. Hence, the former underwent a round of polyploidization event. Polyploidization leads to rapid, extensive genetic and epigenetic changes in the genome. It is associated with many molecular and physiological adjustments and significant gene losses in heterotetraploid cotton after domestication. In tetraploid cotton, the *EXO70* gene family underwent neither large-scale amplification nor reduction according to chromosome location analyses.

Phylogenetic analyses revealed that all eight EXO70 subgroups (EXO70A–EXO70H) are represented in all four cotton species. Genes within the same subgroup had similar structures. Subgroup *EXO70A* consisted of numerous exons, whereas there were relatively fewer in the other seven subgroups. During long-term natural selection, many *EXO70* genes differentiated and evolved to contend with various stressors. A study on Arabidopsis EXO70 demonstrated that recurring gene loss was nonrandom. Genes involved in DNA repair were comparatively more prone to loss, while those implicated in signal transduction and transcription were preferentially retained [49]. Hence, the DNA repair function appears to be absent in the cotton EXO70I subgroup. Other plant species should be examined to determine whether the EXO70I branch is unique to monocots.

### 4.2. Biological Processes Implicating GhEXO70A1-A

EXO70 is one of eight exocyst subunits. It participates in the movement of membrane-related substances and is vital for vesicle transport, cell secretion, growth, division, and other processes [1]. *EXO70* genes are vital to plant growth, development, and metabolism. The study of EXO70 is of great significance to cotton development and metabolism, as this crop is commercially important. Unlike yeast and animals, plants harbor dozens of *EXO70* genes. The latest research shows that *AtEXO70A1* recruits the entire complex to the cytoplasmic membrane by binding to negatively charged phospholipids, providing an important research basis for the study of plant cell polarity and morphogenesis [50]. A functional defect in the *EXO70C2* gene of Arabidopsis affected pollen tube growth and led to male-specific transmission defects [22]. EXO70H4 and PMR4-dependent corpus callosum deposition in trichomes is necessary for cell wall silicification [23]. An *EXO70B1* knockout mutant in Arabidopsis exhibited impaired light-induced stomatal opening [24]. An *OsEXO70L2* mutation caused defects in rice root development [29]. Evidently, the *EXO70* gene has been continuously amplified over the course of plant evolution and displays different physiological functions in various plant tissues and organs.

Transcriptome sequencing indicated that after *GhEXO70A1-A* knockdown in cotton leaf, the expression levels of over 4000 genes significantly changed. Therefore, *GhEXO70A1-A*-gene silencing triggered substantial alterations in the cotton leaf gene expression profile. A DEG function enrichment analysis demonstrated that GhEXO70A1-A function in cotton leaf is strongly correlated with metabolism and especially photosynthesis. Plant metabolism is regulated mainly by various phytohormones, such as auxins, cytokinins, jasmonic acid, abscisic acid, and so on [51,52]. *GhEXO70A1-A* silencing in cotton induces extensive modifications of the metabolic pathways. Thus, *GhEXO70A1-A* loss could affect phytohormone biosynthesis and/or distribution. Studies on Arabidopsis reported that EXO70 is vital for recycling plasma membrane proteins, including the auxin efflux carrier PIN [21,27,53]. The differential functions of various *EXO70* genes in various plant tissues suggest that selective control of specific plant organs is feasible through targeted *EXO70* gene regulation. However, further experimental research in this area must be conducted.

## 5. Conclusions

In the present study, the *EXO70* genes of Asian cotton, Raymond cotton, *G. hirsutum*, and sea island cotton were analyzed by bioinformatics. A total of 165 *EXO70* genes in eight categories were identified. An evolutionary analysis showed that the *EXO70* gene multiplication patterns were nearly identical in upland cotton and sea island cotton, and the same *EXO70* gene types were highly conserved in both varieties. However, *EXO70* gene multiplication in these varieties underwent divergent evolution. Representative research on GhEXO70A1-A revealed that the *EXO70* genes in cotton may be widely involved in metabolism and could also affect phytohormone biosynthesis and/or distribution.

## Figures and Tables

**Figure 1 genes-12-01594-f001:**
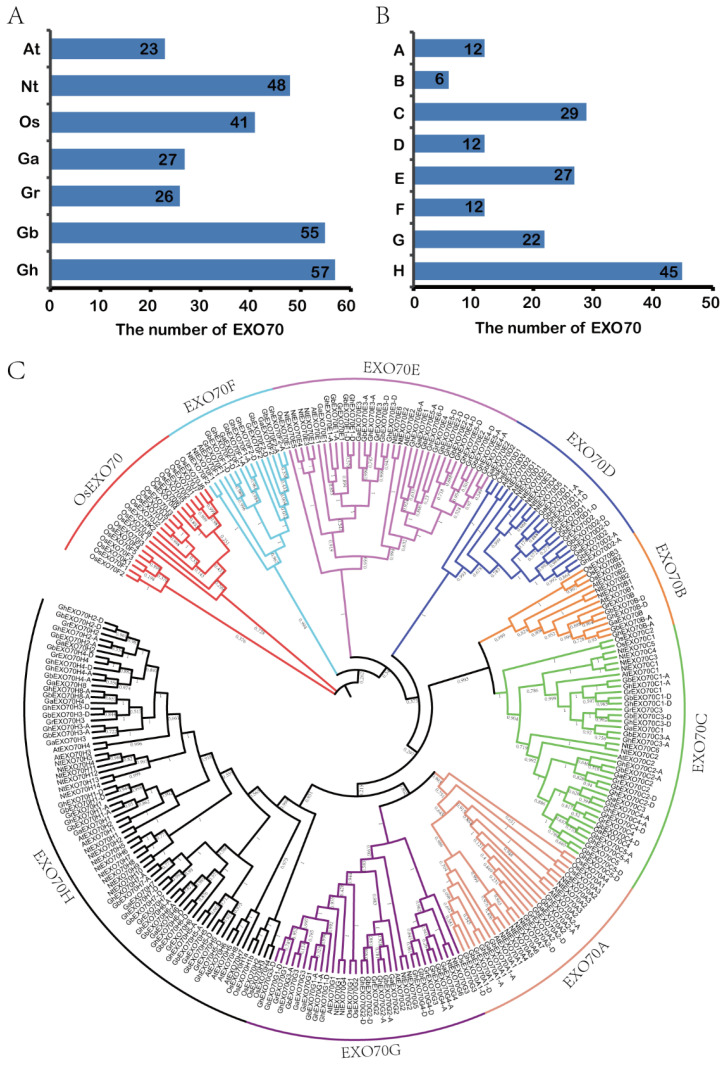
Numbers and phylogenetic relationships of *EXO70* family genes in Arabidopsis, rice, upland cotton, sea island cotton, Asian cotton, Raymond cotton, and tobacco. (**A**) Numbers of *EXO70* family genes in Arabidopsis, rice, upland cotton, sea island cotton, Asian cotton, Raymond cotton, and tobacco. At: *Arabidopsis thaliana*; Nt: *Nicotiana tabacum*; Os: *Oryza sativa*; Ga: *G. arboretum*; Gr: *G. raimondii*; Gb: *G. barbadense*; Gh: *G. hirsutum*. (**B**) Quantitative statistics for each subgroup of the *EXO70* family genes in upland cotton, sea island cotton, Asian cotton, and Raymond cotton. (**C**) Phylogenetic analysis of *EXO70* family genes in Arabidopsis, rice, upland cotton, sea island cotton, Asian cotton, Raymond cotton, and tobacco.

**Figure 2 genes-12-01594-f002:**
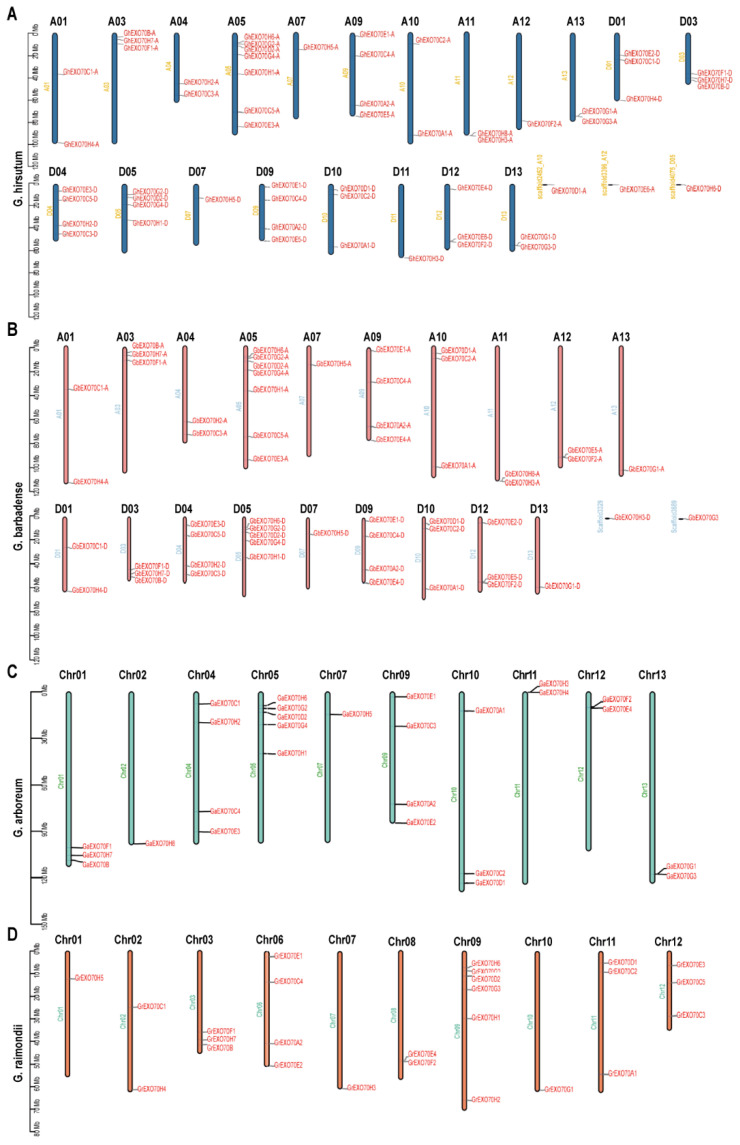
Chromosome distributions of *EXO70* in upland cotton, sea island cotton, Asian cotton, and Raymond cotton. (**A**) Chromosome distribution map of *EXO70* in upland cotton. (**B**) Chromosome distribution map of *EXO70* in sea island cotton. (**C**) Chromosome distribution map of *EXO70* in Asian cotton. (**D**) Chromosome distribution map of *EXO70* in Raymond cotton.

**Figure 3 genes-12-01594-f003:**
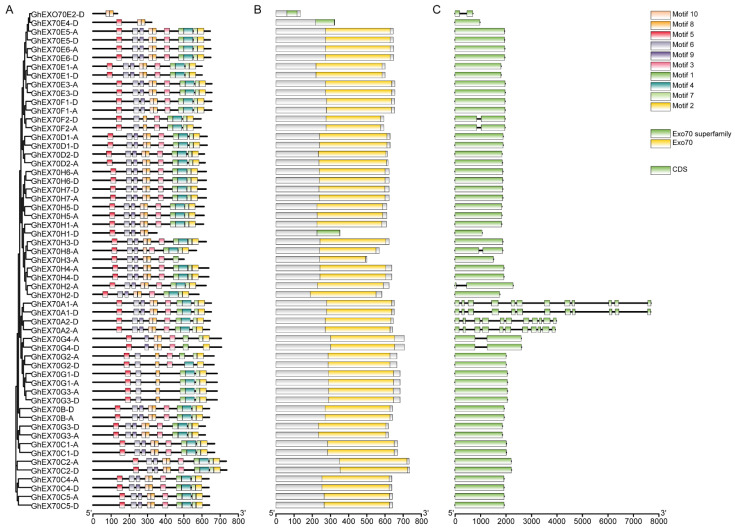
*EXO70* family motif, domain, and gene structure in upland cotton. (**A**) Phylogenetic tree and motif of GhEXO70 proteins. (**B**) The conserved domains in GhEXO70 proteins. (**C**) Gene structure of the *GhEXO70* family.

**Figure 4 genes-12-01594-f004:**
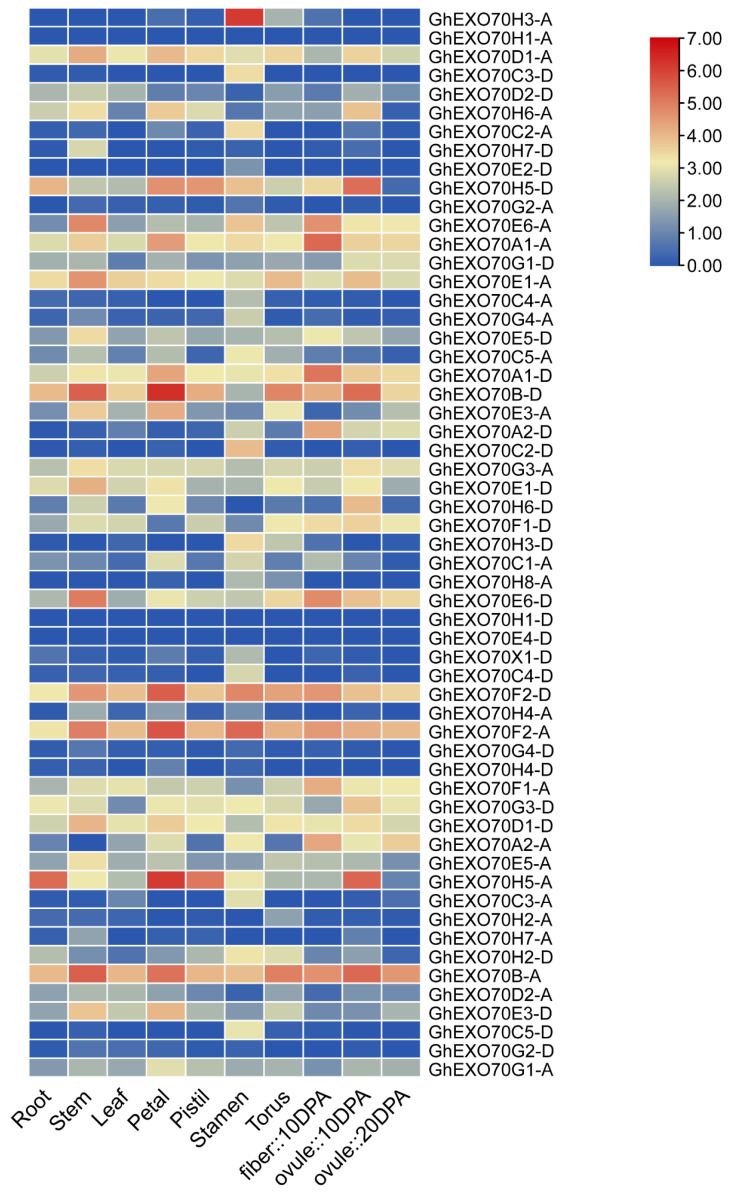
*EXO70* family gene expression patterns in different tissues and organs of upland cotton.

**Figure 5 genes-12-01594-f005:**
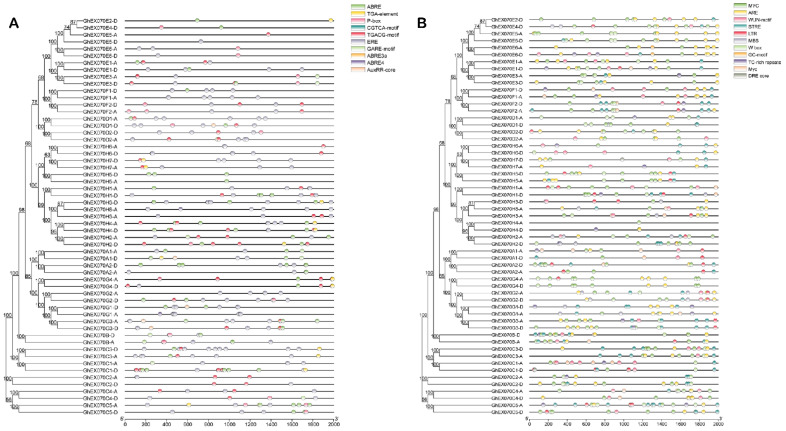
*Cis*-acting elements in *GhEXO70* promoter. (**A**) *Cis*-elements involved in phytohormones were predicted. ABRE: *cis*-acting regulatory element involved in abscisic acid response. AuxRR-core: *cis*-acting regulatory element involved in auxin response. CGTCA-motif: *cis*-acting regulatory element involved in methyl jasmonate (MeJA) response. GARE-motif: gibberellin response element. TGACG-motif: *cis*-acting regulatory element involved in MeJA response. TGA-element: auxin response element. ERE: *cis*-acting ethylene response element. P-box: gibberellin response element. (**B**) Predicted *cis*-elements involved in environmental stress response. GC-motif: enhancer-like elements involved in specific hypoxia induction. LTR: *cis*-acting elements involved in low temperature response. MBS: MYB binding sites related to drought induction. STRE: stress response elements. TC-rich repetitive sequences: *cis*-acting elements involved in defense and stress responses. WUN-motif: wound response elements. MYC: *cis*-acting elements involved in drought stress. W box: *cis*-acting elements involved in sugar metabolism and plant defense signals. DRE core: dehydration response element. ARE: *cis*-acting regulatory element for anaerobic induction.

**Figure 6 genes-12-01594-f006:**
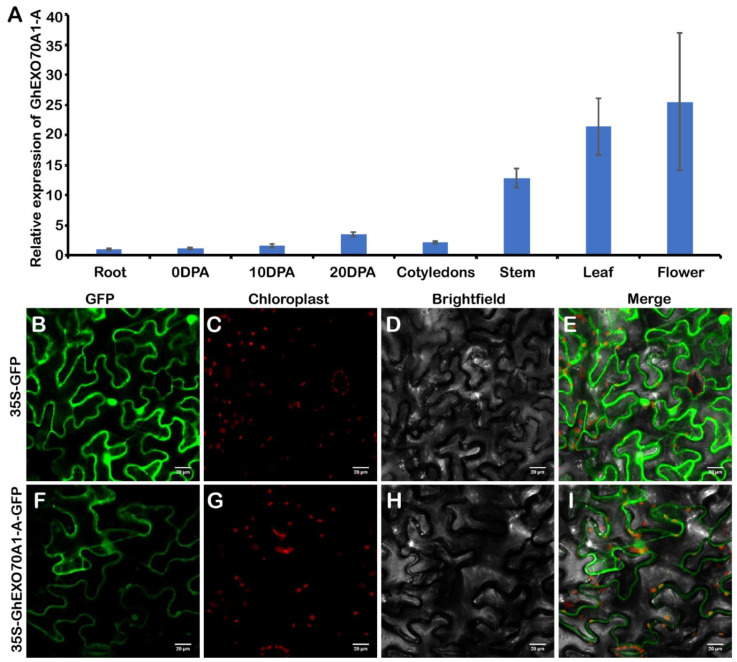
GhEXO70A1-A expression and subcellular localization analyses. (**A**) *GhEXO70A1-A* expression analyses in various upland cotton tissues. (**B**–**I**) Subcellular GhEXO70A1-A localization in tobacco. GFP: green fluorescence. Chloroplast: chloroplast spontaneous red fluorescence. Merge: green and red fluorescence and bright field fusion. (**B**–**E**): 35S-GFP empty vector as control. (**F**–**I**): 35S-GhEXO70A1-A-GFP vector located in plasma membrane. Bar: 10 μm. Data are means ± SD for three replicates.

**Figure 7 genes-12-01594-f007:**
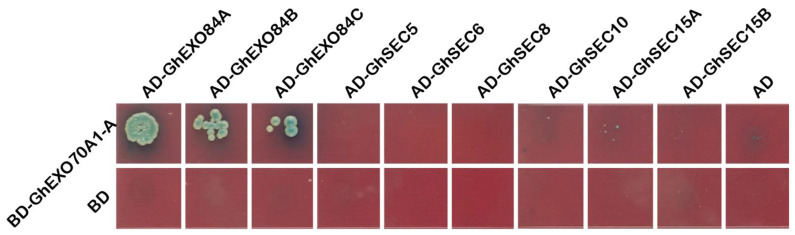
Y2H analysis of interactions among GhEXO70A1-A and other exocyst subunits. GhEXO70A1-A is connected to PGBK-T7 carrier. Other subunits of secretory complex are connected to PGAD-T7 carrier. BD: PGBK-T7 empty vector. AD: PGAD-T7 empty vector.

**Figure 8 genes-12-01594-f008:**
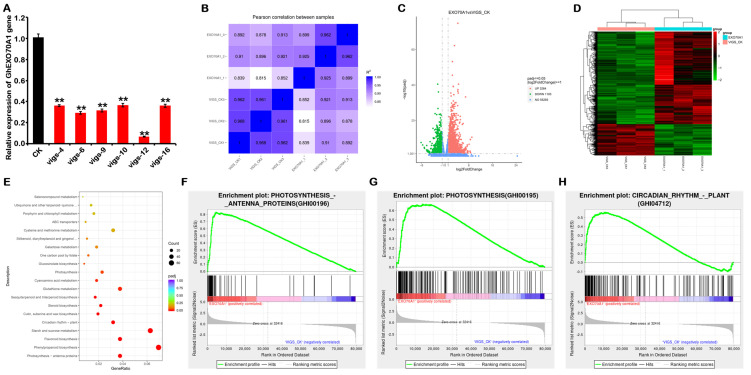
Transcriptome sequencing of differences in cotton transcriptome expression after *GhEXO70A1-A* gene silencing. (**A**) *GhEXO70A1-A* expression levels after virus-induced gene silencing (VIGS). Relative *GhEXO70A1-A* expression levels in plants numbered 4, 6, 9, 10, 12, and 16 significantly decreased. (**B**) Correlation analyses of transcriptome samples. (**C**) Differential gene volcano map in transcriptome. (**D**) Differential gene heat map in transcriptome. (**E**) KEGG functional enrichment dot plot of DEGs. (**F**) GSEA diagram showing changes in photosynthesis antenna proteins after *EXO70A1* gene silencing. (**G**) GSEA diagram showing changes in photosynthetic pathway after *EXO70A1* gene silencing. (H) GSEA diagram showing changes in circadian rhythm pathway after *EXO70A1* gene silencing. Data are means of three replicates ± SD. ** *p* < 0.01.

**Table 1 genes-12-01594-t001:** Nomenclature and analysis of physicochemical properties of *EXO70* family genes in *G. hirsutum*.

Gene ID	Name	Chromosome	Start	End	Exon Number	Protein Length (aa)	Molecular Weight (kDa)	Isoelectric Point	Subcellular Location
Gh_A10G1765	GhEXO70A1-A	A10	92,079,304	92,086,988	12	650	73.432	8.308	Cell membrane
Gh_D10G2039	GhEXO70A1-D	D10	56,152,841	56,160,513	12	650	73.429	8.131	Cell membrane
Gh_A09G1270	GhEXO70A2-A	A09	64,875,878	64,879,803	11	640	72.863	8.901	Cell membrane
Gh_D09G1272	GhEXO70A2-D	D09	39,824,276	39,828,245	11	644	73.467	9.329	Cell membrane, cytoplasm
Gh_A03G0212	GhEXO70B-A	A03	3,226,810	3,228,732	1	640	72.955	4.951	Cell membrane, cytoplasm
Gh_D03G1369	GhEXO70B-D	D03	42,280,554	42,282,476	1	640	72.962	4.914	Cell membrane, cytoplasm
Gh_A01G1064	GhEXO70C1-A	A01	36,760,999	36,763,005	1	668	77.345	8.781	Cell membrane, cytoplasm
Gh_D01G1124	GhEXO70C1-D	D01	23,935,803	23,937,809	1	668	76.941	8.482	Cell membrane, cytoplasm
Gh_A10G0625	GhEXO70C2-A	A10	9,989,979	9,992,177	1	732	84.71	4.555	Nucleus
Gh_D10G0774	GhEXO70C2-D	D10	9,213,316	9,215,523	1	735	85.065	4.537	Nucleus
Gh_A04G0860	GhEXO70C3-A	A04	55,909,662	55,911,515	1	617	70.763	4.975	Cell membrane, cytoplasm
Gh_D04G1359	GhEXO70C3-D	D04	44,235,768	44,237,621	1	617	70.799	5.024	Cell membrane, cytoplasm
Gh_A09G0369	GhEXO70C4-A	A09	20,318,531	20,320,444	1	637	73.612	5.53	Cell membrane, cytoplasm
Gh_D09G0388	GhEXO70C4-D	D09	14,114,761	14,116,674	1	637	73.682	5.35	Cell membrane, cytoplasm
Gh_A05G2929	GhEXO70C5-A	A05	70,962,860	70,964,782	1	640	73.595	6.664	Cell membrane, cytoplasm
Gh_D04G0713	GhEXO70C5-D	D04	14,452,693	14,454,615	1	640	73.566	6.384	Cell membrane, cytoplasm
Gh_A10G2233	GhEXO70D1-A	scaffold2452_A10	2396	4279	1	627	71.136	5.36	Cell membrane, cytoplasm
Gh_D10G0529	GhEXO70D1-D	D10	5,107,684	5,109,567	1	627	71.084	5.278	Cell membrane, cytoplasm
Gh_A05G1157	GhEXO70D2-A	A05	11,706,409	11,708,259	1	616	69.599	5.35	Cell membrane
Gh_D05G1334	GhEXO70D2-D	D05	11,742,012	11,743,850	1	612	69.264	4.972	Cell membrane, cytoplasm
Gh_A09G0090	GhEXO70E1-A	A09	2,303,006	2,304,805	1	599	69.086	4.988	Cell membrane, cytoplasm
Gh_D09G0087	GhEXO70E1-D	D09	2,312,361	2,314,160	1	599	69.308	5.089	Cell membrane, cytoplasm
Gh_D01G1051	GhEXO70E2-D	D01	19,584,687	19,585,379	2	134	15.208	6.674	Cell membrane
Gh_A05G3215	GhEXO70E3-A	A05	84,043,718	84,045,679	1	653	74.613	4.761	Cell membrane
Gh_D04G0392	GhEXO70E3-D	D04	6,206,299	6,208,260	1	653	74.511	4.731	Cell membrane
Gh_D12G0327	GhEXO70E4-D	D12	4,666,459	4,667,427	1	322	36.173	4.875	Cell membrane
Gh_A09G2154	GhEXO70E5-A	A09	74,578,905	74,580,839	1	644	73.351	5.784	Cell membrane, cytoplasm
Gh_D09G2359	GhEXO70E5-D	D09	50,553,799	50,555,733	1	644	73.402	6.24	Cell membrane, cytoplasm
Gh_A12G2651	GhEXO70E6-A	scaffold3396_A12	4667	6610	1	647	73.345	5.788	Cell membrane
Gh_D12G1810	GhEXO70E6-D	D12	50,610,382	50,612,325	1	647	73.122	5.417	Cell membrane
Gh_A03G0449	GhEXO70F1-A	A03	9,703,926	9,705,884	1	652	73.754	4.614	Cell membrane, cytoplasm
Gh_D03G1089	GhEXO70F1-D	D03	36,373,153	36,375,111	1	652	73.79	4.587	Cell membrane
Gh_A12G1712	GhEXO70F2-A	A12	78,884,906	78,886,861	2	593	67.361	4.566	Cell membrane
Gh_D12G1873	GhEXO70F2-D	D12	51,339,965	51,341,920	2	593	67.36	4.589	Cell membrane, cytoplasm
Gh_A13G1576	GhEXO70G1-A	A13	74,625,245	74,627,293	1	682	77.054	8.387	Cell membrane, cytoplasm
Gh_D13G1935	GhEXO70G1-D	D13	54,742,538	54,744,586	1	682	76.932	8.152	Cell membrane, cytoplasm
Gh_A05G0971	GhEXO70G2-A	A05	9,685,126	9,687,123	1	665	74.839	6.629	Cell membrane, nucleus
Gh_D05G1080	GhEXO70G2-D	D05	9,208,639	9,210,636	1	665	74.79	6.457	Cell membrane, nucleus
Gh_A13G1577	GhEXO70G3-A	A13	74,629,925	74,631,973	1	682	77.006	8.013	Cell membrane, cytoplasm
Gh_D13G1936	GhEXO70G3-D	D13	54,747,144	54,749,192	1	682	77.129	8.008	Cell membrane, cytoplasm
Gh_A05G1829	GhEXO70G4-A	A05	19,148,304	19,150,892	2	705	80.991	6.269	Cell membrane, nucleus
Gh_D05G2026	GhEXO70G4-D	D05	18,583,197	18,585,799	2	706	81.133	6.088	Cell membrane
Gh_A05G2577	GhEXO70H1-A	A05	36,616,771	36,618,594	1	607	68.055	7.626	Cell membrane
Gh_D05G2864	GhEXO70H1-D	D05	32,263,469	32,264,524	1	351	39.036	8.216	Cell membrane, nucleus
Gh_A04G0671	GhEXO70H2-A	A04	45,359,837	45,362,111	2	621	70.248	5.721	Cell membrane, cytoplasm
Gh_D04G1136	GhEXO70H2-D	D04	37,216,953	37,218,701	1	582	65.703	5.697	Cell membrane, cytoplasm
Gh_A11G2905	GhEXO70H3-A	A11	92,971,783	92,973,285	1	500	56.067	7.521	Cell membrane, nucleus
Gh_D11G3290	GhEXO70H3-D	D11	65,820,199	65,822,067	1	622	69.832	7.178	Cell membrane, cytoplasm
Gh_A01G1870	GhEXO70H4-A	A01	98,692,379	98,694,289	1	636	71.827	7.493	Cell membrane, cytoplasm
Gh_D01G2127	GhEXO70H4-D	D01	60,327,492	60,329,402	1	636	72.148	7.783	Cell membrane, cytoplasm
Gh_A07G0865	GhEXO70H5-A	A07	15,194,291	15,196,123	1	610	69.021	6.068	Cell membrane, cytoplasm
Gh_D07G0937	GhEXO70H5-D	D07	12,444,178	12,446,010	1	610	68.925	6.316	Cell membrane, cytoplasm
Gh_A05G0839	GhEXO70H6-A	A05	8,379,960	8,381,828	1	622	70.171	5.757	Cell membrane
Gh_D05G3898	GhEXO70H6-D	scaffold4075_D05	141,419	143,287	1	622	70.049	5.361	Cell membrane, cytoplasm
Gh_A03G0316	GhEXO70H7-A	A03	5,680,420	5,682,288	1	622	70.772	5.209	Cell membrane, cytoplasm
Gh_D03G1262	GhEXO70H7-D	D03	40,093,224	40,095,089	1	621	70.603	5.889	Cell membrane, cytoplasm
Gh_A11G2904	GhEXO70H8-A	A11	92,966,319	92,968,183	2	568	63.859	8.556	Cell membrane, cytoplasm

**Table 2 genes-12-01594-t002:** Number of *EXO70s* in each chromosome of different cotton species.

Chromosome	Ga (27)	Gr (26)	Gb (55)		Gh (57)		Total
	A	D	A	D	A	D	
Chr.1	3	1	2	2	2	3	13
Chr.2	1	2	0	0	0	0	3
Chr.3	0	3	3	3	3	3	15
Chr.4	4	0	2	4	2	4	16
Chr.5	5	0	7	5	7	4	28
Chr.6	0	4	0	0	0	0	4
Chr.7	1	1	1	1	1	1	6
Chr.8	0	2	0	0	0	0	2
Chr.9	4	6	4	4	4	4	26
Chr.10	3	1	3	3	2	3	15
Chr.11	2	3	2	0	2	1	10
Chr.12	2	3	2	3	1	3	14
Chr.13	2	0	1	1	2	2	8
total	27	26	27	26	26	28	160
unknown	0	0	1	1	2	1	5

**Table 3 genes-12-01594-t003:** Numbers of *EXO70s* in each subgroup of different cotton varieties.

Subgroup	*G. arboretum*	*G. raimondii*	*G. barbadense*	*G. hirsutum*
A	2	2	4	4
B	1	1	2	2
C	4	5	10	10
D	2	2	4	4
E	4	4	9	10
F	2	2	4	4
G	4	3	7	8
H	8	7	15	15

**Table 4 genes-12-01594-t004:** DEG function pathway enrichment.

KEGG ID	Description	Gene Ratio	Bg Ratio	*p* Value	Up	Down
ghi00196	Photosynthesis antenna proteins	33/877	53/11,853	1.32 × 10^−24^	33	0
ghi00940	Phenylpropanoid biosynthesis	61/877	285/11,853	1.91 × 10^−14^	50	11
ghi00941	Flavonoid biosynthesis	33/877	101/11,853	9.05 × 10^−14^	32	1
ghi00500	Starch and sucrose metabolism	55/877	324/11,853	4.27 × 10^−9^	45	10
ghi04712	Circadian rhythm—plant	28/877	127/11,853	1.26 × 10^−7^	25	3
ghi00073	Cutin, suberine, and wax biosynthesis	17/877	64/11,853	2.56 × 10^−6^	15	2
ghi00100	Steroid biosynthesis	19/877	84/11,853	8.89 × 10^−6^	18	1
ghi00909	Sesquiterpenoid and triterpenoid biosynthesis	15/877	58/11,853	1.41 × 10^−5^	11	4
ghi00480	Glutathione metabolism	33/877	249/11,853	0.00076	20	13
ghi00460	Cyanoamino acid metabolism	16/877	91/11,853	0.00095	16	0
ghi00195	Photosynthesis	20/877	138/11,853	0.002806	20	0
ghi00966	Glucosinolate biosynthesis	7/877	29/11,853	0.004374	1	6
ghi00670	One carbon pool by folate	10/877	56/11,853	0.007353	10	0

## Data Availability

The data presented in this study are available in the Appendix A.

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
