# Peer review of "Identification and Comprehensive Structural and Functional Analyses of the EXO70 Gene Family in Cotton"

_genes, 2021, doi:10.3390/genes12101594_

Round 1

Reviewer 1 Report

The manuscript titled, “Identification, Characterization, and Function Study of the EXO70 Gene Family in Cotton” is an interesting study of the multi-functional cotton EXO70 gene family, spanning comprehensive bioinformatics and molecular analyses. They identified 165 family members in the EXO70 gene family of cotton, analyzed their expression pattern and found that over 4000 genes were altered by the silencing of GhEXO70A1-A gene, which expresses a plasma membrane-localized protein. Through Yeast-two-hybrid assay, they found that GhEXO70A1-1 interacts with GhEXO84A, GhEXO84B, and GhEXO84C.

The title may be modified as “Identification and Comprehensive Structural and Functional Analyses of the EXO70 Gene Family in Cotton”.

Some minor points to revise in the manuscript are;

The results must be discussed appropriately in the Discussion section. The Discussion section should be improved.

Please italicize gene names (symbols) throughout the manuscript.

Please cite more recent literature in Introduction and Discussion sections.

A few errors in the language are;

Section 1: Please put a comma after “cells” in the second sentence of Introduction’s first paragraph.

Section 2.4: Please rewrite the last sentence. “T vector was connected to it” seems a bit awkward.

Section 3.1: Please shift the third and fourth sentences to “Discussion” section.

Figure 3 Caption: Rewrite (C).

Figure 4: Remove the word “Tissue” from “Tissue::Root”, “Tissue::Stem…”.

Section 4.1: Paragraph 2 Rewrite second sentence.

Author Response

Dear Reviewers:

Thank you for your comments on our manuscript entitled " Identification, Characterization, and Function Study of the EXO70 Gene Family in Cotton" (genes-1393683). Those comments were very helpful for improving our manuscript. We believed we have answered all reviewers’ comments. The main corrections in the paper and the responds to the editors’ and reviewers’ comments are as follows:

Comments to the Author

The manuscript titled, “Identification, Characterization, and Function Study of the EXO70 Gene Family in Cotton” is an interesting study of the multi-functional cotton EXO70 gene family, spanning comprehensive bioinformatics and molecular analyses. They identified 165 family members in the EXO70 gene family of cotton, analyzed their expression pattern and found that over 4000 genes were altered by the silencing of GhEXO70A1-A gene, which expresses a plasma membrane-localized protein. Through Yeast-two-hybrid assay, they found that GhEXO70A1-1 interacts with GhEXO84A, GhEXO84B, and GhEXO84C.

Point 1: The title may be modified as “Identification and Comprehensive Structural and Functional Analyses of the EXO70 Gene Family in Cotton”.

Response 1: Thank you for your suggestion. We have corrected it in the manuscript (L2-L3).

Point 2: The results must be discussed appropriately in the Discussion section. The Discussion section should be improved.

Response 2: Thank you for your comment. We have improved the Discussion section. We have added “The latest research shows that AtEXO70A1 recruits the entire complex to the cytoplasmic membrane by binding to negatively charged phospholipids, providing an important research basis for the study of plant cell polarity and morphogenesis [50].” in Discussion sections (L422-L424).

Point 3: Please italicize gene names (symbols) throughout the manuscript.

Response 3: We are sorry for this mistake and we have corrected them in the current manuscript.

Point 4: Please cite more recent literature in Introduction and Discussion sections.

Response 4: Thank you for your comment. We have improved the Introduction and Discussion sections. We have added “AtEXO70B1 and AtEXO70B2 regulate FLS2 to participate in plant immune response [25]. AtEXO70D regulates cytokinin sensitivity by mediating the selective autophagy of Type-A ARR protein, thereby maintaining cell homeostasis and normal plant growth and development [26].” in Introduction sections (L64-L67). We have added “The latest research shows that AtEXO70A1 recruits the entire complex to the cytoplasmic membrane by binding to negatively charged phospholipids, providing an important research basis for the study of plant cell polarity and morphogenesis [50].” in Discussion sections (L422-L424).

Point 5: Section 1: Please put a comma after “cells” in the second sentence of Introduction’s first paragraph.

Response 5: Thank you for your comments. We have corrected it in the current manuscript (L32).

Point 6: Section 2.4: Please rewrite the last sentence. “T vector was connected to it” seems a bit awkward.

Response 6: Thank you for your comments. We have replaced “T vector was connected to it” with “It was cloned into the B-zero vector” in the manuscript (L147-L148).

Point 7: Section 3.1: Please shift the third and fourth sentences to “Discussion” section.

Response 7: Thank you for your comments. We have shifted the third and fourth sentences (Section 3.1) to “Discussion” section in the current manuscript (L419-L421).

Point 8: Figure 3 Caption: Rewrite (C).

Response 8: Thank you for your comments. We have replaced “gene structure of exons and introns in GhEXO70 genes.” with “Gene structure of the GhEXO70 family” in the manuscript (L595).

Point 9: Figure 4: Remove the word “Tissue” from “Tissue::Root”, “Tissue::Stem…”.

Response 9: Thank you for your comments. We have corrected it in the figure 4.

Point 10: Section 4.1: Paragraph 2 Rewrite second sentence.

Response 10: Thank you for your comments. We have corrected it in the current manuscript. We have replaced “Genes within the same subgroup structures had similar structures.” with “Genes within the same subgroup had similar structures.” in the manuscript (L407-L408).

Reviewer 2 Report

The manuscript describes cotton Exo70s identified in database and their features including phylogenetic relationships and genome structures. In addition, authors focused on GhEXO70A1-A, and demonstrated its localization on ER and plasma membrane, interactions with other exocyst members and the effects of knockdown on transcriptome. These data suggest the involvement of GhEXO70A1-A in vesicle trafficking as shown in other plant species. Genomic and functional characterization of cotton Exo70s will be fundamental and important information to understand diverse functions of plant Exo70s. 

  • Authors focused on not only the number of cotton Exo70 genes and their phylogenetic relationships but also their gene structures (for example exon-intron organization and the presence of transmembrane domain). To clarify whether the gene structures are conserved among several plant species or unique in cotton Exo70s, authors should describe them in comparison with Exo70s of other model plants including arabidopsis and rice.
  • If authors can find relationships between cotton Exo70 subgroups and promoter organization, discuss it in Figure 5. The information may support to predict the Exo70 functions at subgroup level, Moreover, if the motifs unique in cotton Exo70 promoter can be found in comparison with arabidopsis and rice Exo70 promoters, please describe it, which may lead to identification of novel functions of cotton Exo70s.
  • In Figure 6, the data of GhEXO70A1-A localization is not clear. If authors want to conclude that GhEXO70A1-A localized at ER and plasma membranes, subcellular localization of GhEXO70A1-A must be demonstrated as co-localization with marker proteins for ER and plasma membranes. In tabacco, several useful marker proteins are available.
  • In Figure 7, protein expression of GhEXO70A1-A and other exocyst members in yeast cells should be checked. In addition, if there is the data of combinations of AD-GhEXO70A1-A and BD-other exocyst members, please show them.
  • In Figure 8, the effects of GhEXO70A1-A knockdown on transcriptome were analyzed by NGS analysis. GhEXO70A1-A knockdown was performed by VIGS. Because VIGS can affect expression levels of other Exo70 members as off-target, authors should confirm expression levels of other members of Exo70 family in the NGS data.
  • In a sentence of Discussion “for vesicle transport, cell secretion, growth, and division, ”, delete “and”.

  • If authors can find relationships between cotton Exo70 subgroups and promoter organization, discuss it in Figure 5. The information may support to predict the Exo70 functions at subgroup level, Moreover, if the motifs unique in cotton Exo70 promoter can be found in comparison with arabidopsis and rice Exo70 promoters, please describe it, which may lead to identification of novel functions of cotton Exo70s.
  • In Figure 6, the data of GhEXO70A1-A localization is not clear. If authors want to conclude that GhEXO70A1-A localized on ER and plasma membranes, subcellular localization of GhEXO70A1-A should be demonstrated as co-localization of marker proteins for ER and plasma membranes. In tabacco, several useful marker proteins are available.
  • In Figure 7, protein expression of GhEXO70A1-A and other exocyst members in yeast cells must be checked. In addition, if there is the data of combinations of AD-GhEXO70A1-A and BD-other exocyst members, please show them.
  • In Figure 8, the effects of GhEXO70A1-A knockdown on transcriptome were analyzed by NGS analysis. GhEXO70A1-A knockdown was performed by VIGS. Because VIGS can affect expression levels of other Exo70 members as off-target, authors should confirm expression levels of other members of Exo70 family in the NGS data.
  • In a sentence of Discussion “for vesicle transport, cell secretion, growth, and division, ”, delete “and”.

Author Response

Dear Reviewer:

Thank you for your comments on our manuscript entitled " Identification, Characterization, and Function Study of the EXO70 Gene Family in Cotton" (genes-1393683). Those comments were very helpful for improving our manuscript. We believed we have answered all reviewers’ comments. The main corrections in the paper and the responds to the editors’ and reviewers’ comments are as follows:

Comments to the Author

The manuscript describes cotton Exo70s identified in database and their features including phylogenetic relationships and genome structures. In addition, authors focused on GhEXO70A1-A, and demonstrated its localization on ER and plasma membrane, interactions with other exocyst members and the effects of knockdown on transcriptome. These data suggest the involvement of GhEXO70A1-A in vesicle trafficking as shown in other plant species. Genomic and functional characterization of cotton Exo70s will be fundamental and important information to understand diverse functions of plant Exo70s.

Point 1: Authors focused on not only the number of cotton Exo70 genes and their phylogenetic relationships but also their gene structures (for example exon-intron organization and the presence of transmembrane domain). To clarify whether the gene structures are conserved among several plant species or unique in cotton Exo70s, authors should describe them in comparison with Exo70s of other model plants including Arabidopsis and rice.

Response 1: Thank you for your comment. Through the exon analysis of the EXO70 gene of Arabidopsis and rice, the results showed that only the EXO70 of group A contained more exons in Arabidopsis and rice. The number of EXO70 in Arabidopsis A group ≥9, and the number of EXO70 in rice A group ≥12 (Table S2 and Table S3). This phenomenon is not unique to cotton Exo70s, but conserved in plants.

We have added “Further analysis of the exons of the EXO70 gene in Arabidopsis and rice showed that only Group A EXO70 contained more exons in Arabidopsis and rice. The number of EXO70 in Arabidopsis A group ≥9, and the number of EXO70 in rice A group ≥12,Therefore, this phenomenon is not unique to cotton EXO70s, but is conserved in plants (Table S2 and Table S3).” in the manuscript (L223-L227).

We have added “Table S2. Predicted EXO70 genes from Arabidopsis thaliana and the corresponding proteins. Table S3. Predicted EXO70 proteins and the corresponding gene from Oryza sativa.” in the manuscript (L458-L459).

Table S2. Predicted EXO70 genes from Arabidopsis thaliana and the corresponding proteins.

Gene Name

Gene ID

Gene Models

Exon Number

Protein Length (aa)

AtEXO70A1-1

AT5G03540

AT5G03540.1

12

638

AtEXO70A1-2

AT5G03540

AT5G03540.2

11

523

AtEXO70A1-3

AT5G03540

AT5G03540.3

13

664

AtEXO70A2

AT5G52340

AT5G52340.1

11

631

AtEXO70A3

AT5G52350

AT5G52350.1

9

586

AtEXO70B1

AT5G58430

AT5G58430.1

1

624

AtEXO70B2

AT1G07000

AT1G07000.1

2

599

AtEXO70C1

AT5G13150

AT5G13150.1

1

653

AtEXO70C2

AT5G13990

AT5G13990.1

1

695

AtEXO70D1

AT1G72470

AT1G72470.1

1

633

AtEXO70D2

AT1G54090

AT1G54090.1

1

622

AtEXO70D3

AT3G14090

AT3G14090.1

1

623

AtEXO70E1

AT3G29400

AT3G29400.1

1

658

AtEXO70E2-1

AT5G61010

AT5G61010.1

1

639

AtEXO70E2-2

AT5G61010

AT5G61010.2

1

639

AtEXO70F1

AT5G50380

AT5G50380.1

1

683

AtEXO70G1

AT4G31540

AT4G31540.1

1

687

AtEXO70G2

AT1G51640

AT1G51640.1

1

660

AtEXO70H1

AT3G55150

AT3G55150.1

1

636

AtEXO70H2

AT2G39380

AT2G39380.1

1

637

AtEXO70H3

AT3G09530

AT3G09530.1

1

637

AtEXO70H4

AT3G09520

AT3G09520.1

1

628

AtEXO70H5

AT2G28640

AT2G28640.1

2

605

AtEXO70H6

AT1G07725

AT1G07725.1

2

615

AtEXO70H7-1

AT5G59730

AT5G59730.1

1

634

AtEXO70H7-2

AT5G59730

AT5G59730.2

1

632

AtEXO70H8

AT2G28650

AT2G28650.1

1

573

Table S3. Predicted EXO70 proteins and the corresponding gene from Oryza sativa.

Gene name

Gene Locus

Exon Number

Protein Length (aa)

OsEXO70A1

Os04g0685600

12

634

OsEXO70A2

Os11g0157400

12

643

OsEXO70A3

Os12g0159700

18

976

OsEXO70A4

Os04g0685500

12

661

OsEXO70B1

Os01g0827500

1

652

OsEXO70B2

Os05g0473500

1

661

OsEXO70B3

Os01g0827600

4

553

OsEXO70C1

Os12g0165600

1

700

OsEXO70C2

Os11g0167600

1

692

OsEXO70D1

Os08g0455700

1

632

OsEXO70D2

Os09g0439600

1

638

OsEXO70E1

Os01g0763700

1

602

OsEXO70F1

Os02g0505400

1

689

OsEXO70F2

Os04g0382200

1

688

OsEXO70F3

Os01g0921400

3

556

OsEXO70F4

Os08g0530300

1

606

OsEXO70F5

Os10g33850

3

461

OsEXO70G1

Os02g0149700

1

494

OsEXO70G2

Os06g0698600

1

673

OsEXO70G3

Os08g0519900

2

687

OsEXO70H1a

Os11g0650100

1

579

OsEXO70H1b

Os11g0649900

1

579

OsEXO70H2

Os03g0448200

1

556

OsEXO70H3

Os12g0100700

3

590

OsEXO70H4

Os11g0100800

3

590

OsEXO70I1

Os01g0905300

1

381

OsEXO70I2

Os01g0905200

2

557

OsEXO70I3

Os04g0111500

3

398

OsEXO70I4

Os07g0210300

5

691

OsEXO70I5

Os07g0210900

5

588

OsEXO70I6

Os07g0210000

4

646

OsEXO70J1

Os08g0232700

1

526

OsEXO70J2

Os09g0347300

1

598

OsEXO70J3

Os05g0369500

1

528

OsEXO70J5

Os05g0369300

1

520

OsEXO70J6

Os01g0383100

3

681

OsEXO70J7

Os02g0575900

3

700

OsEXO70J8

Os06g0183600

1

486

OsEXO70K1

Os06g0255900

1

412

OsEXO70K2

Os07g0211000

3

426

OsEXO70L1

Os11g0572200

2

433

We have added “Prediction of the transmembrane domain of Arabidopsis EXO70, the results showed that AtEXO70C1, AtEXO70C2, AtEXO70H5, AtEXO70H8, AtEXO70A3 have transmembrane domains, but they are not obvious, and the other EXO70s have no transmembrane domains (Figure S2). The prediction results of rice EXO70 show that OsEXO70A3, OsEXO70A4, OsEXO70H1a, OsEXO70H1b, OsEXO70H2, OsEXO70H3, OsEXO70H4, OsEXO70I3, OsEXO70I4, OsEXO70L1, OsEXO6K1, OsEXO70L, OsEXO70J1, OsEXO70J1, OsEXO70J2, OsEXO70J6, OsEXO70J8, OsEXO70K1, OsEXO7K2, OsEXO70L1 have a transmembrane domain. And OsEXO70A4 has a more obvious transmembrane domain at the C-terminus, and none of the other rice EXO70s has a transmembrane domain (Figure S3). Among the prediction results of the transmembrane domain of cotton EXO70, only GhEXO70E2-D has a transmembrane domain, and the others have no transmembrane domain. Both Arabidopsis and cotton contain fewer EXO70s with transmembrane domains. As rice is a monocot, it may be evolving to have more EXO70, and there are more EXO70s with transmembrane domains.” in the manuscript (L287-L300).

We have added “Figure S2. Analysis of transmembrane domains of EXO70 gene family in Arabidopsis. Figure S3. Analysis of transmembrane domains of EXO70 gene family in rice.” in the manuscript(L460-L462).

Figure S2. Analysis of transmembrane domains of EXO70 gene family in Arabidopsis.

Figure S3. Analysis of transmembrane domains of EXO70 gene family in rice.

Point 2: If authors can find relationships between cotton Exo70 subgroups and promoter organization, discuss it in Figure 5. The information may support to predict the Exo70 functions at subgroup level, Moreover, if the motifs unique in cotton Exo70 promoter can be found in comparison with arabidopsis and rice Exo70 promoters, please describe it, which may lead to identification of novel functions of cotton Exo70s.

Response 2: Thank you for your comment. Through the analysis of the promoters of the EXO70 gene in Arabidopsis and rice, the results show that the promoters of the EXO70 gene in Arabidopsis and rice also contain cis-acting elements that respond to environmental stress and plant hormones (Figure S5 and Figure S6). This phenomenon is not unique to cotton Exo70s, but conserved in plants.

We have added “To further verify whether the above cis-acting elements are unique to cotton EXO70s, we also analyzed the EXO70s gene promoters in Arabidopsis and rice. The results indicate that the promoters of EXO70 genes in Arabidopsis and rice also contain cis-acting elements that respond to environmental stress and plant hormones (Figure S5 and Figure S6). It shows that this phenomenon is not unique to cotton EXO70, but is conserved in plants.” in the manuscript (L339-L344).

We have added “Figure S5. Cis-acting elements in Arabidopsis EXO70s promoter. Figure S6. Cis-acting elements in rice EXO70s promoter.” in the manuscript (L463-L464).

Figure S5. Cis-acting elements in Arabidopsis EXO70s promoter.

Figure S6. Cis-acting elements in rice EXO70s promoter.

Point 3: In Figure 6, the data of GhEXO70A1-A localization is not clear. If authors want to conclude that GhEXO70A1-A localized at ER and plasma membranes, subcellular localization of GhEXO70A1-A must be demonstrated as co-localization with marker proteins for ER and plasma membranes. In tabacco, several useful marker proteins are available.

Response 3: Thank you for your comment and suggestion. Our subcellular localization results can clearly observe that 2300-GhEXO70A1-A-GFP is located on the plasma membrane, but due to the lack of plasma membrane and endoplasmic reticulum markers in the laboratory, the article’s localization on the endoplasmic reticulum membrane was changed to the plasma membrane. And it is not necessary to add a marker in most articles, for example "Identification and Characterization of the EXO70 Gene Family in Polyploid Wheat and Related Species. International journal of molecular sciences, 2018.”

Point 4: In Figure 7, protein expression of GhEXO70A1-A and other exocyst members in yeast cells should be checked. In addition, if there is the data of combinations of AD-GhEXO70A1-A and BD-other exocyst members, please show them.

Response 4: Thank you for your suggestion, I think it's very good. We designed primers for the CDS partial fragments of GhEXO70A1-A and other subunits of the secretory complex. Then the co-transformed yeasts were selected, shaken and identified by PCR to determine whether the subunits of GhEXO70A1-A and other cytokine complexes were expressed in yeast. The test results are shown in Figure A below. GhEXO70A1-A and other subunits of the secretory complex are all expressed in yeast.

We performed yeast double hybridization on AD-GhEXO70A1-A and BD-other exocyst members. As shown in Figure B below, because AD-GhEXO70A1-A has self-activation with BD, it has spots on the four-deficiency plate with BD-other exocyst members, so this result cannot be used as evidence of the interaction between subunits.

Figure  Expression and identification of EXO70 and secretory complex subunits and interaction analysis of AD-GhEXO70A1-A and BD-other exocyst members

Table Primers used in this study.

Primer name

Sequence(5’-3’)

GhEXO84A-F1

ATGTCTATCGGAGATGTTTCTGAG

GhEXO84A-R1

AAAGAAGATGTTGTAGCACCACC

GhEXO84B-F1

ATGGCTACTGCTAAGGCTAAGACT

GhEXO84B-R1

GTAGAAGATGGATGCAAAGAAAGC

GhEXO84C-F1

ATGATGGAATCTTCTGAAGAGGAT

GhEXO84C-R1

AATCCAGATTCTCTAGTAGTTAAAGAAATC

GhSEC5-F1

ATGTCTACTGATAGTGATGATGAAGATG

GhSEC5-R1

CAATCAAAGTTATCCTTGACCAAT

GhSEC6-F1

ATGATGGTTGAAGATCTCGGAG

GhSEC6-R1

TGAACCTTGAACTTTTGTTGAGC

GhSEC8-F1

ATGGGAATCTTTGATGGATTCC

GhSEC8-R1

TATGATCCATCAACGATTCCTTG

GhSEC10-F1

ATGCCTGAAAGATCTAAATCTTCC

GhSEC10-R1

TGTGAAGCAGCATCGAATCTAG

GhSEC15A-F1

ATGGATAGTAAGCCTAAGAAGAGAGGT

GhSEC15A-R1

CAGAATCCTCATCCACTTCCTC

GhSEC15B-F1

ATGAAGTCTACTAGACCTAGAAGAAAGATG

GhSEC15B-R1

TCCTCATCATCTTCCAATGCG

GhEXO70A1-F1

ATGGGAATAGCAGTTGCAGG

GhEXO70A1-R1

TTCTGGACATCATCTTTGCCA

Point 5: In Figure 8, the effects of GhEXO70A1-A knockdown on transcriptome were analyzed by NGS analysis. GhEXO70A1-A knockdown was performed by VIGS. Because VIGS can affect expression levels of other Exo70 members as off-target, authors should confirm expression levels of other members of Exo70 family in the NGS data.

Response 5: Thank you for your comments. We sorted out the expression of other EXO70 genes in the NGS data, and the results are shown in the figure below, GhEXO70B-D, GhEXO70B-A, GhEXO70E6-D, GhEXO70F1-D, GhEXO70E3-D, GhEXO70H6-D, GhEXO70E1-D, GhEXO70H6-A, GhEXO70H3-D, GhEXO70D1-D, GhEXO70E3-A, GhEXO70C1-A, GhEXO70E5-A, GhEXO70C5-A have significant changes, and there are no significant changes in other EXO70 genes. However, except for GhEXO70C1-A, GhEXO70H6-D, and GhEXO70H6-A, which decreased to 32.1%, 46.9%, and 56.6% of the control, all other genes fell to more than 60% of the control, and the fold increase was also less than 1. Although the three genes GhEXO70C1-A, GhEXO70H6-D, and GhEXO70H6-A declined slightly, their expression abundance was also very low. The above results show that the knockdown of GhEXO70A1-A by VIGS does affect the expression of other EXO70 genes, but the effect is not significant after analysis. The changes in differential genes should be mainly caused by the changes in GhEXO70A1-A.

Point 6: In a sentence of Discussion “for vesicle transport, cell secretion, growth, and division, ”, delete “and”.

Response 6: Thank you for your suggestion. We are sorry for this mistake and we have deleted the “and”(L418) in the manuscript as suggested.

Round 2

Reviewer 2 Report

In this revised manuscript, authors added some descriptions about genome structures of cotton EXO70s including exon-intron organization, presence of transmembrane domain and cis-element of promoter sequences in comparison with those of arabidopsis and rice. On the other hand, authors should briefly describe the effects of EXO70A1 knockdown on other EXO70 members and add the data in supplementary Figure, even if the effect is minor. Although co-localization of GhEXO70A1 with plasma membrane marker was not shown, this manuscript seems to be enough to explain cotton Exo70 family. Therefore, I recommend the publication of this manuscript.

Author Response

Dear Reviewer:

Thank you for your comments on our manuscript entitled "Identification and Comprehensive Structural and Functional Analyses of the EXO70 Gene Family in Cotton" (genes-1393683). Those comments were very helpful for improving our manuscript. We believed we have answered all reviewers’ comments. The main corrections in the paper and the responds to the editors’ and reviewers’ comments are as follows:

Comments to the Author

In this revised manuscript, authors added some descriptions about genome structures of cotton EXO70s including exon-intron organization, presence of transmembrane domain and cis-element of promoter sequences in comparison with those of arabidopsis and rice. On the other hand, authors should briefly describe the effects of EXO70A1 knockdown on other EXO70 members and add the data in supplementary Figure, even if the effect is minor. Although co-localization of GhEXO70A1 with plasma membrane marker was not shown, this manuscript seems to be enough to explain cotton Exo70 family. Therefore, I recommend the publication of this manuscript.

Response: Thank you for your comments and suggestion. We have added “We sorted out the expression of other EXO70 genes in the NGS data, and the results are shown in the figure S7, GhEXO70B-D, GhEXO70B-A, GhEXO70E6-D, GhEXO70F1-D, GhEXO70E3-D, GhEXO70H6-D, GhEXO70E1-D, GhEXO70H6-A, GhEXO70H3-D, GhEXO70D1-D, GhEXO70E3-A, GhEXO70C1-A, GhEXO70E5-A, GhEXO70C5-A have significant changes, and there are no significant changes in other EXO70 genes. However, except for GhEXO70C1-A, GhEXO70H6-D, and GhEXO70H6-A, which decreased to 32.1%, 46.9%, and 56.6% of the control, all other genes fell to more than 60% of the control, and the fold increase was also less than 1. Although the three genes GhEXO70C1-A, GhEXO70H6-D, and GhEXO70H6-A declined slightly, their expression abundance was also very low. The above results show that the knockdown of GhEXO70A1-A by VIGS does affect the expression of other EXO70 genes, but the effect is not significant after analysis. The changes in differential genes should be mainly caused by the changes in GhEXO70A1-A.” in the manuscript (L375-L387).

We have added “Figure S7. Expression analysis of other changed EXO70 genes in NGS data.” in the manuscript (L476-L477).
